# Relationship between Changes in the Protein Folding Pathway and the Process of Amyloid Formation: The Case of Bovine Carbonic Anhydrase II

**DOI:** 10.3390/ijms232314645

**Published:** 2022-11-24

**Authors:** Bogdan S. Melnik, Natalya S. Katina, Natalya A. Ryabova, Victor V. Marchenkov, Tatiana N. Melnik, Natalya E. Karuzina, Elena V. Nemtseva

**Affiliations:** 1Institute of Protein Research, Russian Academy of Sciences, 142290 Pushchino, Russia; 2Pushchino Branch, Shemyakin-Ovchinnikov Institute of Bioorganic Chemistry, Russian Academy of Sciences, 142290 Pushchino, Russia; 3Biophysics Department, Siberian Federal University, 660041 Krasnoyarsk, Russia; 4Institute of Biophysics, Siberian Branch of Russian Academy of Sciences, 660036 Krasnoyarsk, Russia

**Keywords:** protein folding pathway, amyloidogenesis, bovine carbonic anhydrase, time-resolved fluorescence

## Abstract

Many proteins form amyloid fibrils only under conditions when the probability of transition from a native (structured, densely packed) to an intermediate (labile, destabilized) state is increased. It implies the assumption that some structural intermediates are more convenient for amyloid formation than the others. Hence, if a mutation affects the protein folding pathway, one should expect that this mutation could affect the rate of amyloid formation as well. In the current work, we have compared the effects of amino acid substitutions of bovine carbonic anhydrase II on its unfolding pathway and on its ability to form amyloids at acidic pH and an elevated temperature. Wild-type protein and four mutant forms (L78A, L139A, I208A, and M239A) were studied. We analyzed the change of the protein unfolding pathway by the time-resolved fluorescence technique and the process of amyloid formation by thioflavin T fluorescence assay and electron microscopy. It was revealed that I208A substitution accelerates amyloid formation and affects the structure of the late (molten globule-like)-intermediate state of carbonic anhydrase, whereas the other mutations slow down the growth of amyloids and have either no effect on the unfolding pathway (L78A, L139A) or alter the conformational states arising at the early unfolding stage (M239A).

## 1. Introduction

### 1.1. How Mutations of a Protein Affect Its Ability to Form Amyloids?

Nowadays, the main idea underlying the search for mutations affecting formation and growth of amyloids is focused on finding, in its sequence, an amyloidogenic region, which is a part of a polypeptide chain directly participating in the formation of amyloid structure, and studying amino acid substitutions exactly in this region (for example, [1,2,3,4]). However, the reasons why a protein forms amyloids are more complicated than just the presence of a certain peptide moiety in its amino acid sequence. The majority of proteins do not form amyloid structures during functioning, synthesis, and refolding, even if they have a peptide capable of amyloid formation in their amino acid sequences [5,6]. However, a destabilizing mutation or change of environmental conditions, such as temperature or pH, could lead to the formation of amyloid fibrils [7,8]. This is related to the fact that in most cases, the aggregates or amyloids are formed during the transition of a protein into one of intermediate, weakly structured states. Hence, one can expect that if a mutation leads to formation of novel, «nonconventional» intermediate state(s) of a protein, nonstandard behavior of the protein could appear, i.e., it could be prone to aggregation or amyloid formation. Additionally, vice versa, if a mutation does not affect the protein folding pathway (the sequence of assembly of structural elements during its folding), it most likely will not lead to the appearance of aggregates and amyloids. In the current work, we examined this supposal.

The correlation between the stability of the protein, its intermediate states, and the formation of amyloids has been revealed in many studies (for example [9,10]). In general, the aggregation or formation of amyloids has always been considered to be a consequence of improper protein folding. Therefore, it was assumed that destabilization of the native state of the protein or an increase in the population of intermediate states may cause the increased aggregation [11]. However, in most cases using traditional research approaches, it is not possible to distinguish the two effects of mutation: the destabilization of the “old” intermediate state and the emerging, structurally new intermediate state.

The main problem in examination of the aforementioned supposal is choosing the experimental assay to analyze the protein folding pathway, i.e., the sequence of conformational changes of protein during its folding/unfolding. A number of biophysical methods is usually used for this purpose, such as circular dichroism (CD), infrared spectroscopy, fluorescence, X-ray scattering, sedimentation, etc. Each of them gives the characteristics of the protein at different stages during its folding/unfolding. However, only the analysis of data obtained by various methods allows for the determining of the extent of the effect of a mutation on the protein folding pathway. Therefore, a large-scale study of an extensive number of mutant proteins is an enormous, if not impractical, task.

Our group developed an experimental approach to study the effect of amino acid substitutions on the protein folding pathway, which is based on analysis of the parametric plots [12,13]. These plots reflect the changes of parameters independent on protein concentration and occupancy of intermediate states, but related to protein structure, during protein folding/unfolding. Characteristics of fluorescence decay curves obtained by time-resolved spectroscopy satisfy these criteria. Such an approach is rather novel. Its background and functionality are described in detail in our work [13]. From the experimental point of view, the developed approach turned out to be rather simple, so that a large number of proteins can be studied fast enough. More information on interpreting parametric plots can be found in the Section 1.2.

To test the idea that an amino acid substitutions-changing protein folding pathway should affect the peculiarities of amyloid formation as well, we chose bovine carbonic anhydrase II. It was revealed earlier that the process of folding/unfolding of this protein is complicated and occurs via at least three stages, i.e., via two intermediate states [14,15,16,17]. It was also shown that the parameters obtained from fluorescence decay curves of carbonic anhydrase II solutions in the presence of urea contain information on the main intermediate states of this protein [13]. In addition, it was found that carbonic anhydrase II does not form amyloid structures under normal conditions but does so at changed pH and temperature [18,19].

### 1.2. What Parametric Plots Are and How to Interpret Them

The typical graph representing equilibrium unfolding data for protein reflects dependence of some parameter on the experimental conditions, for example, the intensity of fluorescence or ellipticity on the concentration of a denaturant (Figure 1A,C). 

If protein folding/unfolding occurs via a single stage (i.e., without formation of intermediate states), then the dependences of the different parameters on the denaturant concentration are sigmoidal curves with similar characteristics (midpoints and slopes) (Figure 1A). Moreover, if the curves plotted for various parameters have different characteristics, like is shown in Figure 1C, this indicates the presence of an intermediate state(s) during folding/unfolding of the protein. Such a noncoincidence of transition curves is easy to detect on ideal curves that are strongly shifted relative to each other (like in Figure 1C), but it is much more difficult for raw data with scattered experimental points. The analysis of such curves is simplified by constructing a parametric plot or, in other words, the graph of correlation between parameters. The examples of the dependence of fluorescence intensity on ellipticity for two cases are shown in Figure 1B,D. One can see that the shape of the parametric curve changes significantly if there is an intermediate state during protein folding, i.e., the curve bending on the parametric plot could indicate the presence of an intermediate state. The idea to use such graphs was proposed by Turoverov [20,21,22,23], whose laboratory has investigated a lot of various proteins using this approach. It was shown that the most informative diagram (parametric plot) for determining the number of intermediate states during protein folding/unfolding is that which is created with two spectral parameters [20,21,22].

How to interpret such plots? Each point of the parametric plot is a combination of two spectral parameters related with protein structure. Thus, different points on the plane of this plot correspond to different structural states of the protein, and the curve connecting the points reflects a sequential transition of one structural conformation of the protein to another, for example, when adding different concentrations of denaturant. That is why the curve on the parametric plot can be called the folding/unfolding pathway of the protein, since it denotes the sequence of conformational transformations of the protein during its folding/unfolding.

Moreover, it should be noted separately which parameters are suitable for constructing parametric plots. If it is necessary to determine only the number of intermediate states in a single protein (without comparing with other proteins), then you need to choose parameters that depend both on the structure of the protein and on the population of those intermediate states that we are looking for [23]. If one needs to compare different proteins, it is necessary to use parameters that do not depend on the protein concentration and population of different protein states. This is a very difficult task because most of the optical characteristics (absorption, fluorescence intensity, ellipticity, scattering intensity, etc.) depend on the protein concentration. A few parameters do not have such a dependency. In particular, these are rate constants of protein folding and unfolding in kinetic experiments and fluorescence lifetimes in equilibrium experiments. Plots constructed using such parameters only reflect changes in the structure of the protein but not changes in the population of its states during folding/unfolding. In previous studies, we have shown that the change of two lifetimes of the tryptophan fluorescence of bovine carbonic anhydrase II reflects the formation of two intermediate states of this protein [12].

Additionally, we emphasize that the purpose of parametric plots in this particular work is to monitor changes in the protein structure and not the accumulation of protein intermediate states [13]. Therefore, the use of parameters such as fluorescence intensity or ellipticity for parametric plots is categorically inapplicable. Fluorescence lifetime of the protein is a parameter that does not depend on the population of the intermediate states of the protein and thus can be used for our purpose. Therefore, lifetime-based parametric plots can be interpreted as a pathway of protein structural transformations during its folding. At the same time, it should be taken into account that the use of fluorescence lifetimes (or other parameters that do not depend on protein concentration) does not allow for the distinguishing of intermediate states with the same or similar structure. Nevertheless, if there are several tryptophans in the protein structure and they are located in different parts of the molecule, the use of our method is justified and can provide sufficiently detailed information about the structural changes of the protein during its folding/unfolding. 

The aim of the current study is to compare the effects of amino acid substitutions in protein bovine carbonic anhydrase II on its unfolding pathway and on its ability to form amyloids. To reveal the change of the unfolding pathway, we used parametric plots constructed with two lifetimes of the tryptophan fluorescence of this protein. To analyze the process of amyloid formation of carbonic anhydrase at acidic pH and an elevated temperature, the thioflavin T fluorescence assay and electron microscopy were applied.

## 2. Results and Discussion

### 2.1. Comparison of Unfolding Pathways of Mutant Forms of Carbonic Anhydrase II Using Time-Resolved Fluorescence Technique

Four mutant forms of carbonic anhydrase with substitutions L78A, L139A, I208A, and M239A were studied in the work. Figure 2 shows the protein structure and positions of the substituted amino acid residues. The detailed explanation for why we chose these amino acids to mutate can be found in our earlier works [13,15,17]. Briefly, the main reasons are as follows: these residues are bulky hydrophobic, with the highest number of contacts, and they are located in different structural elements of carbonic anhydrase II (Figure 2). Thus, we hoped that the selected amino acid substitutions would affect different structural elements of the protein and different stages of carbonic anhydrase II unfolding, respectively.

As was revealed earlier [16], the process of folding/unfolding of carbonic anhydrase II is complicated and occurs via formation of two intermediate states:N↔I_1_↔I_2_↔U,(1)
where N is the native state; I_1_—highly structured intermediate state; I_2_—the intermediate state similar to the molten globule; U is the unfolded (denatured) state.

To compare the unfolding pathways of the wild-type carbonic anhydrase and its mutant forms, we measured and analyzed the time-resolved decays of protein fluorescence at different urea concentrations in accordance with the approach described in our previous work [13]. Several parameters can be obtained from the relaxation curves, such as the lifetimes of excited states of tryptophan residues, τ_1_ and τ_2_ and their spectral contributions into fluorescence, A_1_ and A_2_, related to the fluorescence intensity of the studied solutions. Figure 3 shows the curves of equilibrium unfolding plotted using the obtained fluorescence lifetimes (τ_1_, τ_2_) and spectral contributions (A_1_, A_2_) of wild-type carbonic anhydrase and its mutant forms L78A, L139A, I208A, and M239A. 

It can be seen from the indicated midpoints that the structural transitions of the mutants are shifted to the lower urea concentrations compared to the wild-type carbonic anhydrase, i.e., the protein was destabilized by these mutations. In our previous work, we demonstrated that the positions of transition curves plotted by τ_1_ and τ_2_ determined in equilibrium denaturation (Figure 3C,D) are related to the stability of two different intermediate states of carbonic anhydrase, which is in good agreement with the kinetic experiments [12]. Consequently, the shift of curves for mutant forms relative to transitions for wild-type protein in Figure 3C,D could be interpreted as destabilization of each of the intermediate states emerging during folding/unfolding of carbonic anhydrase II caused by mutations. However, the plots in Figure 3 do not allow for concluding that certain substitutions affected the protein in a specific manner.

In our previous work, we demonstrated that revealing the effect of mutations on protein folding pathway requires building parametric plots [13]. Since the parameters τ_1_ and τ_2_ depend neither on protein concentration nor on occupancy of different states of the protein, but they are related to protein structure, each point in the plot of τ_1_ versus τ_2_ characterizes the structural state of the protein but not its occupancy. In other words, the dependency of τ_1_ on τ_2_ plotted for the protein could be interpreted as a protein folding/unfolding pathway (see Section 1.2). 

Figure 4 shows parametric plots of τ_1_ versus τ_2_ for wild-type carbonic anhydrase II and four mutant variants. One can see from Figure 4A that L78A and L139A substitutions only slightly affect the parametric plot and, consequently, do not alter the protein folding pathway. On the contrary, I208A and M239A substitutions apparently influence the shape of the parametric plot (Figure 4B). M239A substitution changes the curve in the area of low urea concentrations (in the range 0–5.6 M). This can be interpreted as the effect on conformational states arising at early unfolding stages (from native state to intermediate state I_1_). In turn, I208A substitution affects conformational states emerging at late stages of the protein unfolding, which leads to changes of the parametric curve under higher urea concentrations of 4.5–8.5 M (Figure 4B).

Thus, one can suppose that if the formation of amyloids of carbonic anhydrase II depends on the structure of molten globule-like intermediate, then I208A could have a specific effect on this process as compared to the other mutant forms [19].

### 2.2. Growth of Amyloid Structures of the Mutant Forms of Carbonic Anhydrase II

Amyloid aggregation of wild-type carbonic anhydrase II and its mutant forms were studied at pH 2.7 and 57 °C. Formation of amyloids by carbonic anhydrase during incubation under such conditions was shown earlier [18,19]. The process of amyloid formation was monitored by thioflavin T fluorescence assay and electron microscopy (Figure 5 and Figure 6).

Figure 5A presents the kinetics of growth of carbonic anhydrase II fibrils detected by thioflavin T fluorescence at 480 nm. Fluorescence spectra of thioflavin T in samples of carbonic anhydrase solutions obtained after reaching the plateau of kinetic curves are shown in Figure 5B. Additionally, this figure displays the fluorescence spectrum of the dye in solution of wild-type protein before incubation, which was identical to the spectra of all studied mutant forms.

One can see in Figure 5A that after 12 days of the incubation, the growth curve of wild-type fibrils reaches plateau, i.e., the growth of fibrillar structures is terminated. Meanwhile, the substitutions L78A, L139A, and M239A decelerate the growth of fibrils, while I208A enhances this process. The curve for I208A in Figure 5A reaches the plateau already on day 8–9.

For each kinetic curve shown in Figure 5A, we calculated lag time and the apparent growth rate constant, using the parameters of sigmoid functions as described in [24]. It allowed for demonstrating that the substitutions L78A and L139A caused the increase in lag time of amyloid growth (Figure 5C), whereas M239A and I208A significantly affect the apparent rate constant of the process (Figure 5D).

Histograms 5C and 5D show that the protein with the I208A substitution differs from other mutated proteins in such parameters as lag time and rate constant.

Electron microscopy studies confirmed that all the investigated proteins formed fibrillar structures after reaching the plateau (Figure 6). However, differences in fibril shape for various mutant proteins can be seen in Figure 6.

Recently, more detailed description of morphological types of these fibrils has been published [25]. It was shown that a distinctive feature of the I208A fibrils is their specific morphology characterized by the lateral protofilaments association and formation of fibril ribbons, while the other mutant variants form worm-like, twisted, or single-long fibrils.

Thus, both thioflavin T binding and electron microscopy data give evidence on formation of fibrillar structures during incubation of wild-type carbonic anhydrase and its mutant forms at pH 2.7 and 57 °C. Taking into account the studies of other groups [18,19], one can suppose that amyloid fibrils were also obtained in our work. 

Analysis of SDS-electrophoresis of carbonic anhydrase II incubated in acidic medium (pH 2.7) at elevated temperature (57 °C) showed that acid hydrolysis of this protein takes place under such conditions. Figure 7 shows SDS-PAGE electrophoregram of wild-type carbonic anhydrase II and its mutant forms after 1-day and 2-day incubation. It can be seen that even after one-day incubation in acidic conditions, only a small fraction of full-size protein remains in the solution, and after two-day incubation, for example, in the case of L78A and L139A mutants, the band of full-size protein is completely absent. 

These results indicated that the observed amyloids are formed not of full-size proteins but of fragments of polypeptide chain derived via hydrolysis of the proteins. The detailed study of this mechanism of amyloid formation of carbonic anhydrase II is presented in recent paper [25]. Briefly, it was revealed that acid hydrolysis during incubation at pH 2.7 and 57 °C leads to the wild-type carbonic anhydrase cleavage into fragments of about 10 kDa and production of free and associated peptides. The latter are homogeneous in molecular weight (about 200 kDa), preserve a pronounced secondary structure of the full-length protein, and act as precursors of amyloid fibrils. Additionally, it was shown that acid hydrolysis of the wild-type carbonic anhydrase is almost complete after 6-day incubation [25]. 

The mass spectrometry method used for the analysis of peptides also confirmed that proteins with different substitutions have a different set of peptides after hydrolysis (the data are not shown). In addition, mass spectrometry does not allow for the determination of the amount of each peptide. Therefore, simple electrophoresis turned out to be more informative for proving that the rate and features of hydrolysis depend on the mutation.

Consequently, the rate-limiting processes in the amyloid growth kinetics shown in Figure 5A are the steps following protein cleavage (i.e., associated peptides formation, fibril elongation). It is noteworthy that a 10 kDa fragment with pronounced secondary structure of the full-length protein could be a β-sheet of carbonic anhydrase, which is the core structural element of this protein, composed with residuals in the range of 85–150 [17].

### 2.3. The Relationship between Mutation Effects on Unfolding Pathway of Carbonic Anhydrase and Its Amyloid Formation

Summing up, we have revealed the following effects of single amino acid substitutions of carbonic anhydrase II on unfolding pathway and amyloids growth:

–Substitution L78A (in β-turn) slightly destabilizes both protein intermediates (Figure 3C,D) but has no effect on the structure of them (Figure 4A); this mutant form exhibits two-times-longer lag time (Figure 5C) and a slightly lower rate of amyloid growth kinetics (Figure 5D) as compared with wild-type protein;–Substitution L139A (in β-sheet) significantly decreases the intermediates’ stability (Figure 3C,D) without changing their structure (Figure 4A); for this mutant variant, a pronounced increase in the lag time (3.6-fold, Figure 5C) and a small decrease in the growth rate (Figure 5D) were found as compared with wild-type protein;–Substitution I208A (in β-turn), in addition to the destabilization of unfolding intermediates of carbonic anhydrase (Figure 3C,D), was found to disturb the structure of the late (molten globule-like, I_2_)-intermediate (Figure 4B); this mutant form demonstrates the fastest amyloid growth kinetics among all studied proteins (Figure 5C,D);–Substitution M239A (in unstructured C-terminal part) causes the lower stability of the protein-unfolding intermediates (Figure 3C,D) and changes the structure of the early-intermediate (I_1_) of carbonic anhydrase (Figure 4B); this mutant variant has a slower amyloids growth kinetics with about a two-times-longer lag time and two-times-lower growth rate (Figure 5C,D) than wild-type protein.

It is notable that the only mutation not localized in the region with a β-structure, M239A, probably led to the growth of amyloids with a similar architecture to wild-type protein (see the fluorescence spectra of thioflavin T in Figure 5B). This result indirectly confirms that both the β-sheet (disturbed by L139A substitution) and two β-turns (affected by L78A and I208A substitutions) of carbonic anhydrase are involved in the formation of amyloid fibrils observed in our study.

Thus, a question arises: why did single amino acid substitutions affect the growth rate and morphology of fibrils? Two proposals could be made: (1) protein hydrolysis occurs, virtually similar, independently on the mutations; thus, the amyloids are formed of similar peptides in all the cases. If the residues selected for substitutions were located in amyloidogenic peptide, it could affect the process of fibril formation; (2) amino acid substitutions affect protein conformation, on which rate and site of polypeptide chain cleavage depends, which, in turn, should affect the peculiarities of fibril formation.

The first hypothesis seems doubtful. The chosen amino acids are not grouped; they are uniformly scattered along the whole polypeptide chain of the protein, so if the fibril formation would depend on mutations in one certain protein region, one should suppose that at least one mutant protein should behave exactly like wild-type protein. However, it is not like that! Each of the mutations did affect both growth rate and morphology of the fibrils (Figure 5 and Figure 6). 

The following chain of events could be reasonably proposed: (I) mutations affect the structure of intermediate states of carbonic anhydrase; (II) rate and features of hydrolysis of polypeptide chain (cleavage site position) depend on the structure of intermediate states; (III) peculiarities of hydrolysis (length and composition of peptides), in turn, affect the rate and morphology of amyloid aggregation.

It is not difficult to assume that some mutations lead to a change in the structure of intermediate states of a protein. Which mutations do affect and which do not can be determined by different experimental methods, but we believe that for tryptophan-containing proteins, the simplest approach is to measure time-resolved fluorescence and to plot parametric plots. It is reasonable that the conditions for the manifestation of a mutation effect are related to the conditions for the formation of an intermediate state of the protein. Some of the intermediate states are realized in acidic pH, so the mutations affecting the structure of them also influence protein hydrolysis under these conditions. The growth of amyloids is just an indicator of the fact that the mutation has changed the intermediate state of the protein (for carbonic anhydrase, this leads to protein cleavage in acidic pH, depending on the mutation).

The fact that the mutations in different parts of carbonic anhydrase affect the stability of different structural elements of this protein and, consequently, conformation of different intermediate states of the protein, was demonstrated in our previous work [17]. This gives an indirect confirmation to the stated proposal. Furthermore, different peptides could form during acid hydrolysis of mutant variants of the proteins. For example, it is clear that, only in case of wild-type protein, the intensity of bands after one-day and two-day incubation changes only slightly; in other cases, it is not like that (Figure 7). This can give evidence on different peptide composition in studied protein samples. For example, in Figure 7, we indicated by arrows the bands specific for I208A and M239A mutant proteins which, apparently, are absent in the other protein preparations. Of course, electrophoresis is a rough experimental method, but even this method shows that the hydrolysis of the studied proteins occurs with a different rate and polypeptide chain cleavage occurs in different sites.

Figure 8A shows the CD spectra of wild-type carbonic anhydrase and mutant proteins at pH 8 and pH 2.7 before incubation and, respectively, before their acid hydrolysis. Figure 8B shows SDS electrophoresis of the same samples stained with Kumasi. It can be seen that the proteins are intact before incubation and that their mobility is the same and corresponds to a protein with a molecular weight of approximately 30 kDa.

It can be seen from the CD spectra that: (1) at pH 8, all proteins are the same; that is, mutations did not significantly affect the structure of carbonic anhydrase under native conditions; (2) the CD spectra are of low intensity and do not make it possible to distinguish the features of different structural states of the protein upon partial unfolding of BCAB. It can be seen that the spectra of all mutant proteins at pH 2.7 are very similar; minor differences between the spectra cannot be correctly interpreted. (3) At pH 2.7, proteins are structured; they have a lot of the secondary structure. Calculations using the program BeStSel (https://bestsel.elte.hu, access date: 2 November 2022) show that carbonic anhydrase in pH 8 alpha = 44%, beta = 30%, and turn = 26%, and in pH 2.7, alpha = 28%, beta = 30%, and other = 32%.

## 3. Materials and Methods

### 3.1. Isolation and Purification of Carbonic Anhydrase II and Its Mutant Forms

Recombinant vectors encoding wild-type bovine carbonic anhydrase II (BCA II) and its mutant forms L78A, L139A, I208A, and M239A were obtained in our earlier studies [12,13,15,16,17]. Isolation and purification of BCA II and its mutant forms from overproducing the *E. coli* strain were made according to the technique described earlier [26,27]. Homogeneity of protein preparations was controlled by gel electrophoresis in native and denaturing conditions. The native state of the recombinant carbonic anhydrase II was controlled by restoration of esterase activity of the protein [26]. 

Protein concentration was determined by 280 nm UV light absorption using Cary 100 spectrophotometer (Agilent Technologies, Santa Clara, CA USA). The extinction 1.87 was taken for BCA II and its mutant forms [26,27].

### 3.2. Preparation of Carbonic Anhydrase II Amyloids

Liophilized protein powder was dissolved in 50 mM phosphate–citrate buffer (pH 2.7); the solution was centrifuged at 120,000× *g* for 30 min to remove all the formed associates on Beckman 100 ultracentrifuge (Beckman Coulter, Brea, CA, USA). The amyloids were prepared by incubation of 4 mg/mL protein solution at 57 °C in TS-100 thermal shaker (Biosan, Latvia) at 300 rpm mixing rate.

### 3.3. SDS–Polyacrylamide Gel Electrophoresis (SDS-PAGE)

To study protein hydrolysis, electrophoresis under denaturing conditions was carried out according to a conventional technique [28]. PAAG with gradient porosity (9–15%) was used, and the proteins were stained with Coomassie Blue G250.

### 3.4. Electron Microscopy

For electron microscopy, the protein solution (0.1 mg/mL) was applied to copper wire mesh with formvar film and contrasted by 1% uranyl acetate solution. The micrographs were obtained on a transmission electron microscope JEM 1400 EX (Jeol Ltd, Akishima, Japan) at 80 kV acceleration voltage.

### 3.5. Fluorescence Thioflavin T Measurements

Fluorescence spectra of thioflavin T were recorded on Cary Eclipse spectrofluorimeter (Agilent Technologies, Santa Clara, CA, USA) at 25 °C using a cuvette with optical path length 3 mm. Protein concentration used to measure fluorescence was 4 μM (0.12 mg/mL); dye concentration was 25 μM. Excitation wavelength was 450 nm; spectra were registered in the range 460–600 nm. Kinetic curves were obtained by measuring fluorescence intensity at 480 nm.

To estimate the lag time, t_lag_, and apparent rate constant, k, of amyloid growth, we used the approach described in [24]. Namely, we approximated each kinetic trace with sigmoidal function and used the fitted parameters to calculate t_lag_ and k.

### 3.6. Steady-State and Time-Resolved Fluorescence Measurements

To construct parametric plots, fluorescence of the BCA II (wild-type protein and mutant forms) was measured at 20 °C in Tris-HCl buffer (20 mM, pH 8) containing increasing concentrations of urea (0–8 M). Steady-state fluorescence of the proteins at the concentration of 0.16 mg/mL was measured using Fluorolog 3-22 (Horiba Jobin Yvon, Edison, NJ, USA) spectrofluorometer. The excitation wavelength was 295 nm. The emission spectra were recorded in the range of 300–450 nm. All spectra were corrected for the background signal and for the inner filter effect.

Time-resolved fluorescence measurements were performed using DeltaHub timing module of Fluorolog 3-22 (Horiba Jobin Yvon, USA). A NanoLED pulsed diode (296 nm, pulse duration of ~1.2 ns) was applied as an excitation source. Time-resolved fluorescence decays were collected in the range of 305–417 nm with a step of 4 nm; time resolution was 0.027 ns/channel. The instrument response function (IRF) was measured using a Ludox solution detecting the emission at the excitation wavelength.

To recover the fluorescence decay parameters (amplitudes and lifetimes), the global analysis approach was performed using the DAS6 software (Horiba Jobin Yvon, USA). The set of decays collected for each sample was modeled by iterative reconvolution with three exponential components (see [12] for details). As a result, the time-resolved fluorescence decay at the wavelength λ was described as:$$I_{\lambda }\left(t\right)=\overset{3}{\underset{i=1}{\sum }}\alpha_{i}^{\lambda }\exp\left(-t/\tau_{i}\right),$$
where $\tau_{i}$ is the lifetime and $\alpha_{i}^{\lambda }$ is the amplitude of the *i*-component. 

The fractional fluorescence intensity $f_{i}^{\lambda }$ of each lifetime component was determined as
$$f_{i}^{\lambda }=\frac{\alpha_{i}^{\lambda }\tau_{i}}{\sum_{j=1}^{N}\alpha_{j}^{\lambda }\tau_{j}}.$$

Then, the normalized contribution of the *i*-component to the steady-state fluorescence intensity $A_{i}$ was calculated as
$$A_{i}=\frac{\sum_{\lambda }I_{\mathrm{ss}}^{\lambda }f_{i}^{\lambda }}{\sum_{\lambda }I_{\mathrm{ss}}^{\lambda }},$$
where $I_{\mathrm{ss}}^{\lambda }$ is the steady-state fluorescence intensity at wavelength λ. The contribution of the shortest lifetime (τ_3_ < 0.25 ns) was found to be small (A_3_ < 0.1), which was the reason for excluding it from consideration.

Using the urea-induced transition curves obtained from the change of the time-resolved fluorescence parameters (P), the fraction of the protein, which completes the transition, was calculated by the following equation:$$B_{after}\left(\mathrm{urea}\right)=1-\frac{P\left(\mathrm{urea}\right)-P_{after}\left(\mathrm{urea}\right)}{P_{before}\left(\mathrm{urea}\right)-P_{after}\left(\mathrm{urea}\right)}.$$

Here, *P*(urea) is a parameter value under certain urea concentration; *P_before_*(urea) and *P_after_*(urea) are the baseline values of the parameter for the protein states before and after the transition. *P_before_*(urea) and *P_after_*(urea) were calculated using equations obtained from the linear fitting of the data outside the transition area.

The dependences of *B*_after_ on urea concentration were fitted by the Boltzmann function available in the Origin 8.0 software (Origin, OriginLab Corp., MD, USA).
(2)$$B_{after}=\frac{B_{start}-B_{end}}{1+e^{\left(\left[\mathrm{Urea}\right]-{\left[\mathrm{Urea}\right]}_{m}\right)/w}}+B_{end},$$
where *B*_start_ and *B_end_* are the initial and final values of B, which are close to 1 and 0, respectively, [Urea] is urea concentration, [Urea]_m_ is transition midpoint, and w refers to the transition width.

Since carbonic anhydrase denaturation is not a two-state process, the transition midpoints [Urea]_m_ could be considered to be apparent parameters. 

## 4. Conclusions and Proposals

The hypothesis which we tested in the current work is that the formation of amyloids or specific aggregates of large globular proteins depends, first of all, not on the presence of amyloidogenic peptides in their polypeptide chains, but on conformational states that occur during protein folding/unfolding. The theoretical confirmation of this hypothesis could be found in the works demonstrating, by bioinformatics methods, that potential amyloidogenic peptides are present in virtually any protein [5,6]. Meanwhile, it is known that not all the proteins form amyloids during their synthesis and functioning. 

In this work, we studied four mutant variants of bovine carbonic anhydrase II with single amino acid substitutions in different parts of the sequence. All the substitutions (L78A, L139A, I208A, and M239A) reduce hydrophobicity of the residues. If amyloid growth depends only on amino acid composition of the protein, i.e., on the presence of amyloidogenic peptide, we could expect that such substitutions in protein amyloidogenic regions would lead to the deceleration of aggregation [24]. At the same time, substitution outside the amyloidogenic region would have zero effect on the process of amyloid formation. Indeed, obtained data reveal the decrease in aggregation rate for all studied proteins except the one with substitution I208A (Figure 5C,D).

This mutation (I208A) accelerated amyloid growth. It is difficult to explain such an effect by changes in the amyloidogenic region, but it could be explained if we suppose that different mutations affected the structure of intermediate states of carbonic anhydrase, on which, in turn, the following process of aggregation or amyloid formation depends. The support to this suggestion was obtained by a time-resolved fluorescence method which showed that only I208A substitution affects the structure of late (molten globule-like)-intermediate states of carbonic anhydrase. 

Furthermore, an important result of the current work is the successful test of an experimental approach allowing for the revelation of the mutual relation between the effect of mutation on protein folding pathways and on amyloidogenesis. Taking into account these data and our previous results [12,13], we can conclude that using time-resolved fluorescence allows for the exploring of a large number of mutant protein forms and for the revealing of the main features of the effect of these mutations on protein folding pathways.

Further, we will focus on the possibility of investigating other proteins by the method used in this work. Now, we are testing our approach on several proteins and their mutant forms. Preliminarily, we concluded that if protein contains more than one tryptophan residue in its structure, then this method is certainly applicable. The time-resolved fluorescence technique appears to be much more informative than the conventional (steady state) fluorescence method and often gives results that can be obtained only in kinetic experiments [12].

## Figures and Tables

**Figure 1 ijms-23-14645-f001:**
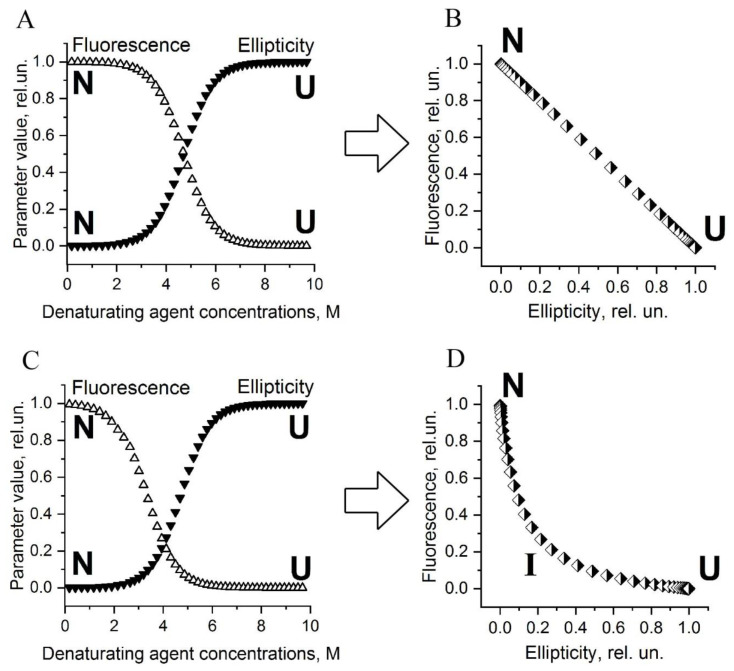
Schematic representation of typical transition curves used to study the equilibrium unfolding of protein (**A**,**C**) and corresponding parametric plots (**B**,**D**). Transition curve reflects the dependence of some experimental characteristic of protein sample on denaturant concentration (**A**,**C**). In the parametric plots (**B**,**D**), each point is a combination of two experimental characteristics of sample measured at a certain denaturant concentration. (**A**,**B**) are the graphs for a protein which fold/unfold without the formation of intermediate states. (**C**,**D**) are the graphs for a protein which fold/unfold through the formation of one intermediate state. N, I, U denote parts of the graphs characterizing the native, intermediate, and unfolded states of the protein, respectively.

**Figure 2 ijms-23-14645-f002:**
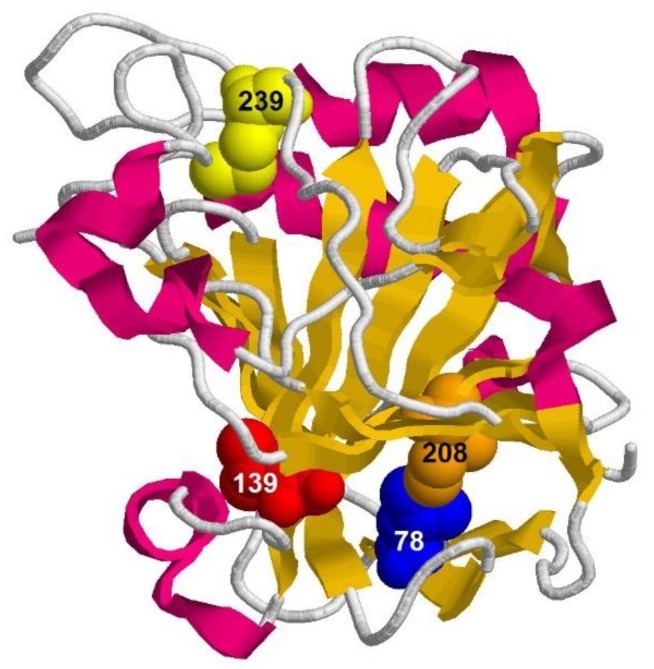
Three-dimensional model of bovine carbonic anhydrase II (PDB: 1v9e). The substituted amino acid residues are shown in space-filled representation. Amino acids located in the structural elements of the protein, which are formed at different stages of folding of carbonic anhydrase, were selected for replacement. As we showed earlier [17], the protein structure in the region of I208 and L139 is formed first, and the loop in the region of L78 is formed at the last stage of carbonic anhydrase folding.

**Figure 3 ijms-23-14645-f003:**
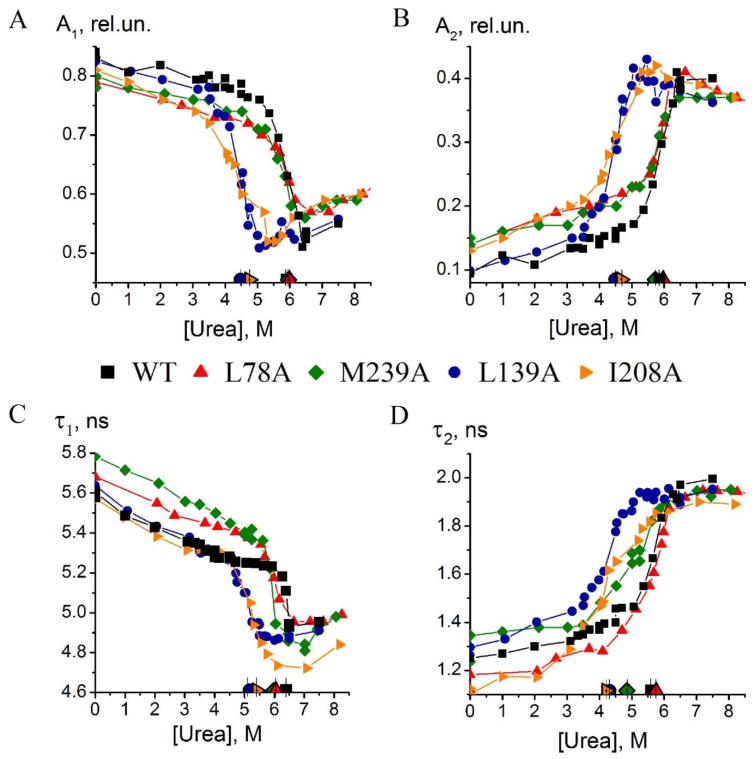
Dependencies of spectral contributions A_1_, A_2_ (**A**,**B**) and lifetimes τ_1_, τ_2_ (**C**,**D**) calculated from fluorescence decay curves on urea concentrations for carbonic anhydrase II solutions: wild-type (black), L78A (red), L139A (blue), I208A (orange), M239A (green). The solid line serves to guide the eyes. Markers with ticks near the abscissa axis refer to the apparent transition midpoints of the corresponding curves.

**Figure 4 ijms-23-14645-f004:**
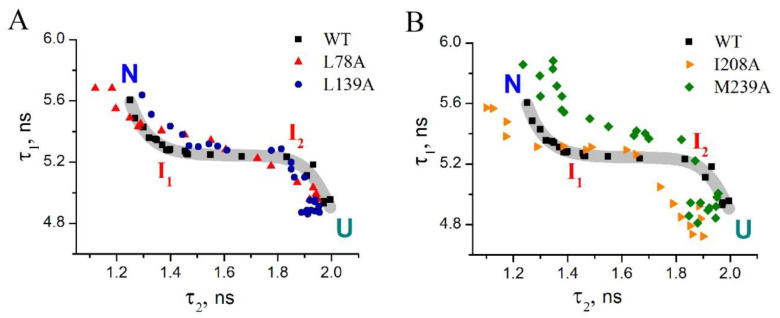
Mutual dependency of two fluorescence lifetimes, τ_1_ and τ_2_, during urea-induced unfolding for carbonic anhydrase: wild-type (black), L78A (red), L139A (blue), I208A(orange), M239A (green). The smoothed dependency for wild-type protein is shown by a thick gray line. Bends on the curve indicate the presence of intermediate states during the protein unfolding. N, U, I_1_, I_2_ indicate the areas of the parametric plot for wild-type protein that correspond to the native, unfolded, and intermediate states, respectively. On panel (**A**)—the proteins whose unfolding pathway does not differ from each other, on panel (**B**)—proteins whose unfolding pathway is different.

**Figure 5 ijms-23-14645-f005:**
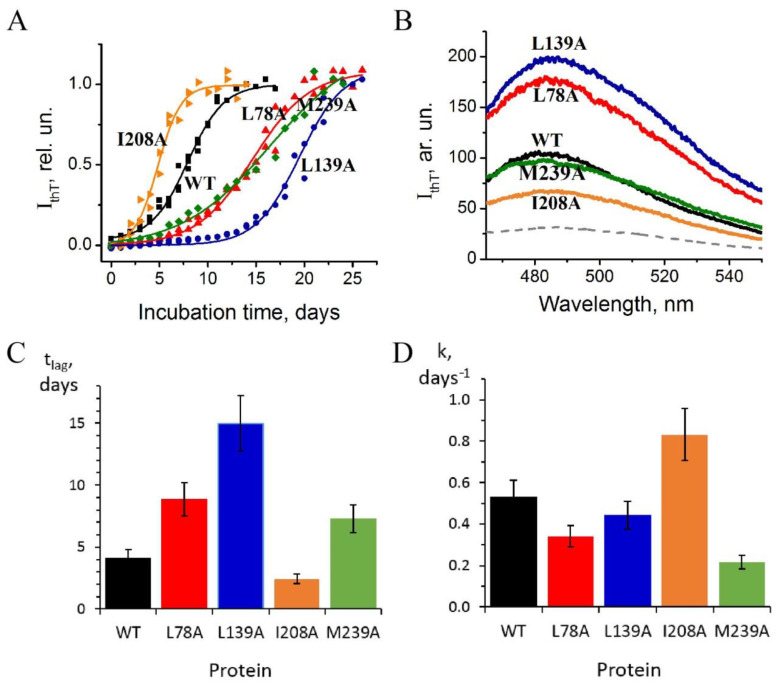
(**A**) Dependence of intensity of thioflavin T fluorescence at 480 nm on time of incubation at pH 2.7, 57 °C, in the samples of carbonic anhydrase: wild-type (black), L78A (red), L139A (blue), I208A (orange), M239A (green). The signal is normalized to the plateau value. Solid lines are the approximations with the Boltzmann function. (**B**) Fluorescence spectra of thioflavin T in the samples of carbonic anhydrase solutions after reaching the plateau of kinetic curves shown in A; gray dashed line indicates the spectrum of thioflavin T in the wild-type carbonic anhydrase before protein incubation. (**C**) Lag times, t_lag_, and (**D**) growth rate constants, k, for the kinetic traces shown in (**A**) for the calculation, the method published in [24].

**Figure 6 ijms-23-14645-f006:**
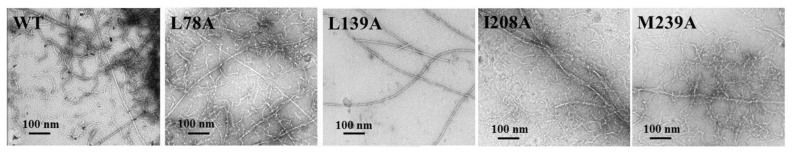
Electron micrographs of amyloid aggregates of wild-type carbonic anhydrase and its mutant forms after reaching the plateau of kinetic curves (for WT—17 days of incubation, for L78A—23 days, for L139A and M239A—25 days, for I208A—14 days).

**Figure 7 ijms-23-14645-f007:**
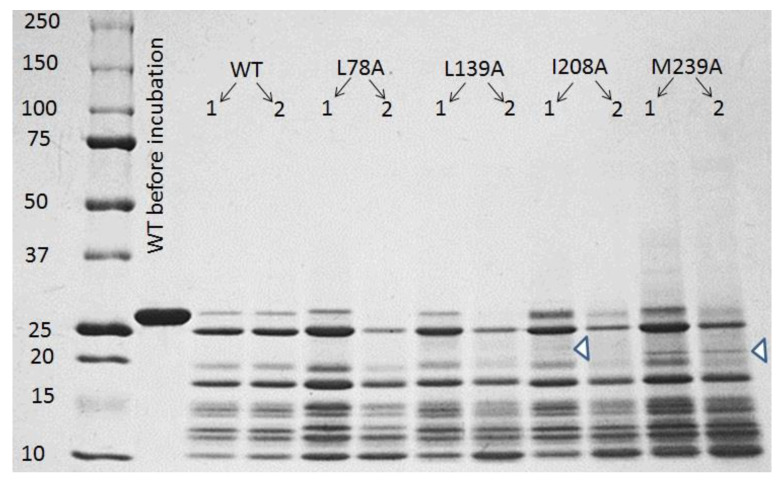
SDS-PAGE electrophoregrams of carbonic anhydrase solutions, wild-type and L78A, L139A, I208A, M239A mutant forms, incubated at pH 2.7, 57 °C for one (1) and two (2) days. Left column: molecular weight markers. The second column from the left: wild-type carbonic anhydrase before the incubation. The proteins were stained with Coomassie Blue G250.

**Figure 8 ijms-23-14645-f008:**
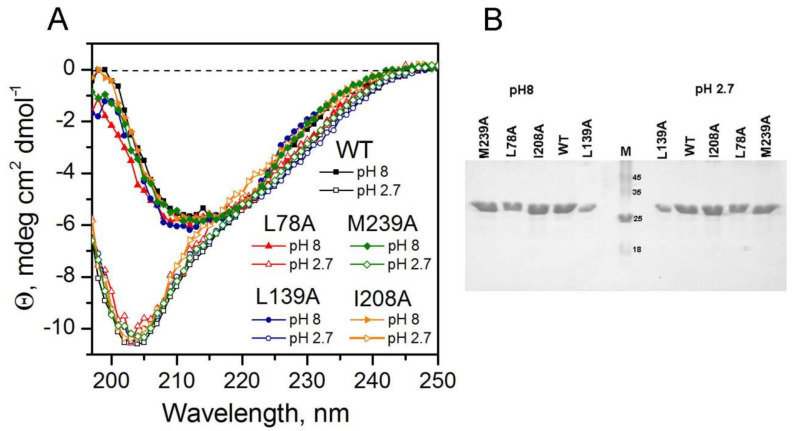
(**A**) CD spectra samples of carbonic anhydrase: wild-type (black), L78A (red), L139A (blue), I208A (orange), M239A (green) before incubation at pH8 (filled characters) and pH 2.7 (empty characters), 20 °C. (**B**) SDS-PAGE electrophoregrams of the same samples before incubation at 57 °C. M—molecular weight markers.

## Data Availability

The data presented in this study are available on request from the corresponding author.

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
