# Peer review of "Relationship between Changes in the Protein Folding Pathway and the Process of Amyloid Formation: The Case of Bovine Carbonic Anhydrase II"

_ijms, 2022, doi:10.3390/ijms232314645_

Round 1

Reviewer 1 Report

This study by Melnik et al. examines the hypothesis of a direct correlation between the effect of mutations in the primary sequence of a protein on its folding pathway and fibrillogenesis. In general, the manuscript presents important results that are potentially of interest to the readers of the journal.

The authors are kindly asked to address some shortcomings before it can be accepted.

Major:

It should be noted that the phase diagram method was developed to analyze denaturation curves and detect intermediate states (including hidden ones) that accompany the processes of protein unfolding and folding (Burstein, E. A. Mol. Biol. (Mosk) 1971, 5, 214-225; Kuznetsova, I.M.; Turoverov, K.K.; Uversky, V.N. J.Proteome.Res. 2004, 3, 485–494). The method is based on the parametric representation of two independent extensive characteristics of the protein system under denaturing influences. Such characteristics describe the system quantitatively, i.e. linearly depend on the proportions of all states of the protein molecule. Thus, the method of phase diagrams is theoretically and mathematically substantiated. Intensive parameters, including the position of the fluorescence maximum, the value of parameter A, fluorescence anisotropy, lifetime, etc, which qualitatively characterize the system, are absolutely inapplicable in parametric analysis. The method proposed by the authors for carbonic anhydrase II is based on the empirical observation of the similarity of the true phase diagrams with the dependence between the lifetime components and the authors' assumption that the later one reflects the properties of two intermediate states of the protein. There is no reason to generalize this method to other proteins.

Minor:

1)       The symbols denoting data related to carbonic anhydrase II wt and its mutant variants in figures 3, 4 and 5A and in their captions do not correspond to each other. I recommend to remove the symbols from the captions – the color symbols in the figures are more informative.

2)       The significance of the differences for the data presented in figures 5C and 5D should be estimated.

3)       The references should be cheeked through the text. A number of references appear to be in error: reference 27 on page 10, line 329 and page 13, line 461, reference 29 on page 11, lines 367 and 373.

Author Response

Reviewer 1

This study by Melnik et al. examines the hypothesis of a direct correlation between the effect of mutations in the primary sequence of a protein on its folding pathway and fibrillogenesis. In general, the manuscript presents important results that are potentially of interest to the readers of the journal.

The authors are kindly asked to address some shortcomings before it can be accepted.

Major:

It should be noted that the phase diagram method was developed to analyze denaturation curves and detect intermediate states (including hidden ones) that accompany the processes of protein unfolding and folding (Burstein, E. A. Mol. Biol. (Mosk) 1971, 5, 214-225; Kuznetsova, I.M.; Turoverov, K.K.; Uversky, V.N. J.Proteome.Res. 2004, 3, 485–494). The method is based on the parametric representation of two independent extensive characteristics of the protein system under denaturing influences. Such characteristics describe the system quantitatively, i.e. linearly depend on the proportions of all states of the protein molecule. Thus, the method of phase diagrams is theoretically and mathematically substantiated. Intensive parameters, including the position of the fluorescence maximum, the value of parameter A, fluorescence anisotropy, lifetime, etc, which qualitatively characterize the system, are absolutely inapplicable in parametric analysis. The method proposed by the authors for carbonic anhydrase II is based on the empirical observation of the similarity of the true phase diagrams with the dependence between the lifetime components and the authors' assumption that the later one reflects the properties of two intermediate states of the protein. There is no reason to generalize this method to other proteins.

Our response:    Thank you for your comment. We certainly know the works of Turoverov (references 20-23) in which parametric graphs are developed, their application is substantiated and many proteins are used to study. We understand perfectly why extensive characteristics are chosen for the analysis of intermediate states and also use them, but not in this work. To analyze the protein folding pathway (by folding pathway we mean structural changes in the protein but not the accumulation of its different states), we developed and substantiated a new variation of parametric plots which uses only parameters (characteristics) that do not depend on the population of protein intermediate states (refs. 12, 13). I want to emphasize that the lifetimes of tryptophans in an excited state are parameters that do not depend on either the protein concentration or the population of the intermediate states, and were chosen intentionally. It is not possible to compare the folding pathways of mutant proteins using extensive characteristics!

We have described our method in detail in our article (13. Nemtseva, et al. Experimental approach to study the effect of mutations on the protein folding pathway. PLoS One 2019, 14, e0210361), we refer to it in the text. We understand that our approach is not standard. The common purpose is to detect the accumulation of intermediate states (i.e. to monitor their population), but our purpose is to monitor the change in protein structure by methods that do not "see" the accumulation of different protein states (do not depend on population and concentration). It was important for us to distinguish cases where a mutation affects the order of formation of structural elements from cases where a mutation only affects the population of a state.

We have added additional explanations to the text of the article:

Additionally, we emphasize that the purpose of parametric plots in this particular work is to monitor changes in the protein structure, and not the accumulation of protein intermediate states [13]. Therefore, the use of parameters such as fluorescence intensity or ellipticity for parametric plots is categorically inapplicable. Fluorescence lifetime of the protein is a parameter that does not depend on the population of the intermediate states of the protein and thus can be used for our purpose. Therefore, lifetime-based parametric plots can be interpreted as a pathway of protein structural transformations during its folding. At the same time, it should be taken into account that the use of fluorescence lifetimes (or other parameters that do not depend on protein concentration) does not allow distinguishing intermediate states with the same or similar structure. Nevertheless, if there are several tryptophans in the protein structure and they are located in different parts of the molecule, the use of our method is justified and can provide sufficiently detailed information about the structural changes of the protein during its folding/unfolding.

Reviewer 1    Minor:

  • The symbols denoting data related to carbonic anhydrase II wt and its mutant variants in figures 3, 4 and 5A and in their captions do not correspond to each other. I recommend to remove the symbols from the captions – the color symbols in the figures are more informative.

Our response:

We have corrected the figure captions (for figures 3-5) in accordance with this recommendation.

Fig. 3. Dependencies of spectral contributions A1, A2 (A, B) and lifetimes t1, t2 (C, D) calculated from fluorescence decay curves on urea concentrations for carbonic anhydrase II solutions: wild type (black), L78A (red), L139A (blue), I208A (orange), M239A (green). Solid lines are the approximation of the transitions with the Boltzmann function. Markers with ticks near the abscissa axis refer to the transition midpoints of the corresponding curves.

Fig. 4. Mutual dependency of two fluorescence lifetimes, t1 and t2, during urea-induced unfolding for carbonic anhydrase: wild type (black), L78A (red), L139A (blue), I208A (orange), M239A (green). The smoothed dependency for wild-type protein is shown by thick gray line. Bends on the curve indicate the presence of intermediate states during the protein unfolding. N, U, I1, I2 indicate the areas of the parametric plot for wild-type protein that correspond to the native, unfolded and intermediate states of the wild-type protein, respectively.

Fig. 5. (A) Dependence of intensity of thioflavin T fluorescence at 480 nm on time of incubation at рН 2.7, 57°C, in the samples of carbonic anhydrase: wild type (black), L78A (red), L139A (blue), I208A (orange), M239A (green). The signal is normalized to the plateau value. Solid lines are the approximations with the Boltzmann function.  (B) Fluorescence spectra of thioflavin T in the samples of carbonic anhydrase solutions after reaching the plateau of kinetic curves shown in A; gray dashed line indicates the spectrum of thioflavin T in the wild-type carbonic anhydrase before protein incubation. (C) Lag times, tlag, and (D) growth rate constants, k, for the kinetic traces shown in (A) as obtained from the parameters of fitted curves [27].

Reviewer 1   

  • The significance of the differences for the data presented in figures 5C and 5D should be estimated.

Our response:  The plots in Figures 5C and 5D were not discussed in detail because they were added to show that the I208A mutant protein differs from other proteins in two more parameters. We have added explanations to the text of the article.

Histograms 5C and 5D show that the protein with the I208A substitution differs from other mutated proteins in such parameters as lag time and rate constant.

Reviewer 1   

3)       The references should be cheeked through the text. A number of references appear to be in error: reference 27 on page 10, line 329 and page 13, line 461, reference 29 on page 11, lines 367 and 373.

Our response:   Thank you very much! Indeed, there was a mistake in the references. We have corrected it.

Reviewer 2 Report

This manuscript can be divided into two distinct sections, analysis of stability of single mutants of BCA and aggregation at low pH. The first one, wherein four mutations have been carried out, were analysed based on the parametric plot that has been developed by the authors as is visible from the referred publications. Based on the authors, the lifetime analyses is free of bias as it only reports on the protein structure. Two of the mutants did not affect the protein folding pathway while I208A and M239A did as per the deviation from the parametric plot of the wild-type protein. Incubation at pH 2.7 at a high temperature led to aggregation of the protein. However, under those conditions, the protein is seen to hydrolyse to different fragments which subsequently aggregate. The authors have then correlated the aggregation kinetics of the fragments to that available from the lifetime plots. The latter part I somehow cannot get myself to agree with. The fact that the mutation will affect the structural intermediates is ok. Beyond this, such mutations affecting protein hydrolysis, rate of the same and hence having an effect on the final morphology is highly speculative.

Have independent experiments (apart from gel) been carried out to see what the involved fragments are after hydrolysis? Since aggregation goes on for an extended period of time, some insights into the aggregating fragments should have been obtained. Once the protein is fragmented, the aggregation profile should have little dependence on the starting intact protein, unless proved otherwise through parallel experiments. It might happen that the fragments themselves are bringing about the observed changes and any relation to the intact protein at present is purely coincidental. Had the results been on an intact BCA, I would not have had such issues. Also, as a suggestion, why does one have to go to low pH? To avoid the complexity of hydrolysis, can the aggregation not be carried out at a certain urea concentration which might provide the same starting point (prior to hydrolysis) as that of pH 2.7?

Finally understanding protein folding pathways and aggregation has a strong structure dependence and the parametric plot can definitely be an added way of finding out which intermediate gets affected or not. Such plots without any basic structural information does not provide much insights.

·         I would have expected that the authors to have carried out the CD measurements as a function of urea for all the mutants and show that indeed would show the benefit of having this parametric plot. No matter how much one tried to be convincing with a parametric plot, the progressive dissolution in structural components can only be obtained through the CD spectra. Hence CD spectra are necessary to have along with the time-resolved fluorescence data.

·         The CD spectra become more important at pH 2.7 wherein the protein undergoes hydrolysis. However, no connection to the secondary structures of the fragments can be seen in this manuscript.

·         The authors have mentioned that they have used the Boltzmann function to fit the fluorescence parameters? I hope they mean the trend as shown in Figure 3. They have also mentioned about obtaining transition midpoints from the fits. Why was the Boltzmann function used? Should one rather not have used the linear extrapolation method (LEM) as is often used assuming a two-state process. Next, when the parametric plot (Figure 4) shows the presence of at least two intermediates, then what is the significance of a transition mid-point?

·         Why do the urea axes start from 2, why not from 0? One needs to know the lifetime components of native BSA.

·         On page 8, the authors mention that the lifetime parameters do not depend upon protein states but provide information on the protein structure --- this was not clear to me; will different protein states not be having different conformations/structure?

·         The pulse duration of the LED is 1.2 ns. Then what is the IRF. I guess it would be close to 1.4 ns. In that case, I have ample reservation with regards to the A2 values and tau2 components as the initial tau2 values are very close to the IRF, that is, almost overlapping with the IRF.

·         In Figure 2, I did not understand the meaning of the statement ‘Structural elements formed at different stages of protein folding are displayed in different colors’; the authors should expand on the same; which color corresponds to what stage?.

Author Response

Reviewer 2:  

Comments and Suggestions for Authors

This manuscript can be divided into two distinct sections, analysis of stability of single mutants of BCA and aggregation at low pH. The first one, wherein four mutations have been carried out, were analysed based on the parametric plot that has been developed by the authors as is visible from the referred publications. Based on the authors, the lifetime analyses is free of bias as it only reports on the protein structure. Two of the mutants did not affect the protein folding pathway while I208A and M239A did as per the deviation from the parametric plot of the wild-type protein. Incubation at pH 2.7 at a high temperature led to aggregation of the protein. However, under those conditions, the protein is seen to hydrolyse to different fragments which subsequently aggregate. The authors have then correlated the aggregation kinetics of the fragments to that available from the lifetime plots. The latter part I somehow cannot get myself to agree with. The fact that the mutation will affect the structural intermediates is ok. Beyond this, such mutations affecting protein hydrolysis, rate of the same and hence having an effect on the final morphology is highly speculative.

Have independent experiments (apart from gel) been carried out to see what the involved fragments are after hydrolysis? Since aggregation goes on for an extended period of time, some insights into the aggregating fragments should have been obtained. Once the protein is fragmented, the aggregation profile should have little dependence on the starting intact protein, unless proved otherwise through parallel experiments. It might happen that the fragments themselves are bringing about the observed changes and any relation to the intact protein at present is purely coincidental. Had the results been on an intact BCA, I would not have had such issues.

Our response:     Thank you for reading our article carefully!

I understand your explanations/doubts. You're right. Indeed, we do not fully understand the process of the mutation influence, since the results obtained differed greatly from those expected.

Our initial assumption was the same as you expressed - after protein fragmentation, the aggregation process should proceed in the same way for all mutant proteins. We hoped that only one of the proteins would differ in amyloid morphology, and then only if the mutation ideally affected the region that forms amyloids. Since only one amino acid was replaced in each of the mutant proteins, no strong differences could be expected. However, there was a significant difference between the mutant proteins.

Therefore, we think that the result can be explained by the following sequence of events:

The mutation affects the structure of the protein, which in turn affects the site of hydrolysis of the protein chain, the features of hydrolysis affect the size and composition of protein fragments, which subsequently aggregate and form different amyloids. Such a logical chain more or less explains the experimental result. We couldn't find another explanation.

Hydrolysis was repeated many times, electrophoresis was performed at different times after incubation, many samples for electron microscopy were obtained - the main result is that different mutations affect protein hydrolysis in different ways. Apparently, this is the reason for the different rates of aggregation and different morphology of amyloids. (For the same mutation, the results of the experiments are repeated and coincide)

We tried to analyze by mass spectrometry the peptides that were obtained as a result of hydrolysis. We hoped that the set of peptides would cover the same regions of the protein despite different mutations.

Unfortunately, the result was different, and the situation was not cleared up. For each mutant protein, a different set of peptides was obtained. In addition, mass spectrometry does not allow the determination of the amount of each peptide (at least our mass spectrometer). Therefore, simple electrophoresis turned out to be more informative for proving that the rate and features of hydrolysis depend on the mutation.

The mass spectrometry method used for the analysis of peptides also confirmed that proteins with different substitutions have a different set of peptides after hydrolysis.

Reviewer 2:    Also, as a suggestion, why does one have to go to low pH? To avoid the complexity of hydrolysis, can the aggregation not be carried out at a certain urea concentration which might provide the same starting point (prior to hydrolysis) as that of pH 2.7?

Our response:     As we have established earlier, urea prevents aggregation, so it definitely cannot be used. In addition, the original purpose was to grow amyloids under the same conditions as other authors [18]. Now, in our opinion, even the fact that in subsequent studies it is necessary to more accurately check how the formation of amyloids occurs - from the whole protein or as a result of fragments aggregation, is of value.

Reviewer 2:     Finally understanding protein folding pathways and aggregation has a strong structure dependence and the parametric plot can definitely be an added way of finding out which intermediate gets affected or not. Such plots without any basic structural information does not provide much insights.

Our response:     This is true, but we approached the problem from the other side. We have done many experiments using CD, fluorescence, microcalorimetry, etc. [ref] but they don't give an understanding of the folding pathway. All these methods are integral and changes, for example, in the CD spectrum, do not clarify the situation. Therefore, the idea was to use time-resolved fluorescence since the local environment of tryptophans strongly influences the excited state lifetime. We will certainly continue our research. For example, we created mutant proteins in which we replaced each of the tryptophans, this will allow us to understand their contribution to lifetimes and correlate different parts of the curves on parametric plots with the structural parts of the protein. Unfortunately, this part of the work is progressing slowly and not easily. I hope this will be our next article.

Reviewer 2:     I would have expected that the authors to have carried out the CD measurements as a function of urea for all the mutants and show that indeed would show the benefit of having this parametric plot. No matter how much one tried to be convincing with a parametric plot, the progressive dissolution in structural components can only be obtained through the CD spectra. Hence CD spectra are necessary to have along with the time-resolved fluorescence data. ·

The CD spectra become more important at pH 2.7 wherein the protein undergoes hydrolysis. However, no connection to the secondary structures of the fragments can be seen in this manuscript.

Our response:     In our previous work [12-13], we published data on the unfolding of mutant forms of carbonic anhydrase, studies were performed by the fluorescence method, and kinetic experiments were carried out. CD studies for this protein are not informative because it has a low intensity CD spectrum (see Fig. 8), so conformational changes in the protein have minor effect on the spectrum, while impurities, such as urea, can greatly increase the noisiness of the curve. As a result, reliable spectra cannot be obtained and interpreted.

The advice to compare the CD spectra of proteins in native conditions and at pH 2.7 is very good, thank you. We have added a picture with CD spectra and explanations to it.

Figure 8A shows the CD spectra of wild-type carbonic anhydrase and mutant proteins at pH8 and pH2.7 before incubation and, respectively, before their acid hydrolysis. Figure 8B shows SDS electrophoresis of the same samples stained with Kumasi. It can be seen that the proteins are intact before incubation, their mobility is the same and corresponds to a protein with a molecular weight of approximately 30 kDa.

It can be seen from the CD spectra that: 1) At pH8, all proteins are the same, that is, mutations did not significantly affect the structure of carbonic anhydrase under native conditions; 2) The CD spectra are of low intensity and do not make it possible to distinguish the features of different structural states of the protein upon partial unfolding of BCAB. It can be seen that the spectra of all mutant proteins at pH 2.7 are very similar, minor differences between the spectra cannot be correctly interpreted. 3) At pH 2.7, proteins are structured, they have a lot of secondary structure. Calculations using the program BeStSel (https://bestsel.elte.hu) show that carbonic anhydrase in pH8 alpha=44 beta=30 turn=26 in pH2.7 alpha=28 beta=30 other=32.

Reviewer 2:    The authors have mentioned that they have used the Boltzmann function to fit the fluorescence parameters? I hope they mean the trend as shown in Figure 3. They have also mentioned about obtaining transition midpoints from the fits. Why was the Boltzmann function used? Should one rather not have used the linear extrapolation method (LEM) as is often used assuming a two-state process.

Our response:     Indeed, the transitions revealed from dependences tau([Urea]) and A([Urea]) were fitted by Boltzmann function (sigmoidal curve) to obtain the transition midpoints and to compare these parameters for mutant forms and wild-type protein. Application of the Boltzmann function to “raw” experimental data allows determining the transition midpoint more directly than linear extrapolation method (LEM). However, the midpoint values obtained by these two methods for one data set should be close to each other, and they are (as we checked). Transition midpoints are shown in Figure 3 to demonstrate that lifetime components tau1 and tau2 are more informative parameters during BCA denaturation than their contributions A1 and A2: dependences tau1([Urea]) and tau2([Urea]) give distinct transition midpoints for one protein, reflecting two successive stages of protein unfolding, while A1([Urea]) and A2([Urea]) bare the same information (because A1+A2=1).

Our previous investigations [ref. 12-13 in the manuscript] revealed that for carbonic anhydrase II the changes of lifetime components tau1 and tau2 in equilibrium denaturation with urea correspond to the two different unfolding stages of this protein determined by kinetic methods. In other words, we concluded that tau1 is insensitive o the second stage of unfolding, whereas tau2 does not change during the first stage. With this assumption, we can use Boltzmann equation to the transitions appeared in tau1 and tau2 dependencies.

We have added the Boltzmann equation to Methods section of the manuscript.

In the text ¯:

The urea-induced transition curves obtained from the change of the time-resolved fluorescence parameters (P) were fitted by the Boltzmann function, available in the Origin 8.0 software (OriginLab Corp.):

   ,

where Pstart and Pend are initial and final values of the parameter, respectively, [Urea] is urea concentration, [Urea]m is transition midpoint, w refers to the transition width.

The transition midpoints [Urea]m were taken from this fitting for consideration.

Reviewer 2:     Next, when the parametric plot (Figure 4) shows the presence of at least two intermediates, then what is the significance of a transition mid-point?

Our response:     It is difficult to show the transition mid-point on parametric graphs (Fig. 4), it is not convenient, and there is no reason to.

A feature of parametric plots is that their shape does not depend on the experimental conditions. This has advantages and disadvantages. The disadvantage is that the mid-point of the transitions between different intermediate states on such plots cannot be determined. The transition mid-point is better defined in plots like Figure 3. The advantages of parametric plots can be explained by the example of WT and L139A proteins. In Figure 3, the curves for WT and L139A proteins are different, while in Figure 4 they are identical. Thereby, the stability of proteins is different (they have different transition mid-points in Fig. 3), but the sequence of their conformational states during unfolding is the same (the shape of the curves in Fig. 4). That is, proteins have the same folding pathway, despite different stability. This, in fact, is the purpose of our parametric plot.

Reviewer 2:    Why do the urea axes start from 2, why not from 0? One needs to know the lifetime components of native BSA.

Our response:     We have corrected the figure 3 in accordance with this comment (extended the range to 0-8.5 М). Previously we tried to focus on the range where the transitions appeared.

Reviewer 2:   On page 8, the authors mention that the lifetime parameters do not depend upon protein states but provide information on the protein structure --- this was not clear to me; will different protein states not be having different conformations/structure?

Our response:     Lifetime parameters do not depend on the experimental conditions but depend on the conformation/structure of the protein! If we talk about one protein, then this statement seems wrong, because by changing the experimental conditions (for example, the concentration of urea), we will change the conformation of the protein, but if we compare different proteins, then the meaning becomes clear. For example, it may be that the destabilized mutant protein under native conditions is already in the same conformation as the WT protein when 2M urea is added.

Reviewer 2:    The pulse duration of the LED is 1.2 ns. Then what is the IRF. I guess it would be close to 1.4 ns. In that case, I have ample reservation with regards to the A2 values and tau2 components as the initial tau2 values are very close to the IRF, that is, almost overlapping with the IRF.

Our response:     We performed a reconvolution analysis to recover the fluorescence decay parameters, using special software DAS6 (Horiba). For this, we measured the profile of the IRF using Ludox scattering solution and analyzed it with the fluorescence decays.

According to investigations [1-2] and, as a consequence, to the practical guide from our instrument’s manufacturer [3], such a data processing allow measuring fluorescence lifetimes as short as 1/10 of the full width at half maximum of IRF. So, we could register the values of tau2 (which are >0.95 ns) and A2 quite confidently with pulse duration of the LED as long as 1.2 ns. Examples of applying the same excitation source and instrument to estimate the fluorescence lifetimes of about 1 ns could be readily found [4-5]. We have added more details about this issue into Methods section of the manuscript.

References:

  1. Birch, D. J., & Imhof, R. E. (2002). Time-domain fluorescence spectroscopy using time-correlated single-photon counting. In Topics in fluorescence spectroscopy(pp. 1-95). Springer, Boston, MA.
  2. Cova, S., Ghioni, M., & Zappa, F. (1991). Optimum amplification of microchannel‐plate photomultiplier pulses for picosecond photon timing. Review of scientific instruments62(11), 2596-2601.
  3. https://www.horiba.com/fileadmin/uploads/Scientific/Downloads/UserArea/Fluorescence/Manuals/IBH-TCSPC-Guide.pdf
  4. Maurya, J. K., Mir, M. U. H., Singh, U. K., Maurya, N., Dohare, N., Patel, S., ... & Patel, R. (2015). Molecular investigation of the interaction between ionic liquid type gemini surfactant and lysozyme: A spectroscopic and computational approach. Biopolymers103(7), 406-415.
  5. Voicescu, M., Heinrich, M., & Hellwig, P. (2009). Steady-state and time resolved fluorescence analysis on tyrosine–histidine model compounds. Journal of fluorescence19(2), 257-266.

changes in the text ¯:

Time-resolved fluorescence measurements were performed using DeltaHub timing module of Fluorolog 3-22 (Horiba Jobin Yvon, USA). A NanoLED pulsed diode (296 nm, the pulse duration of ~ 1.2 ns) was applied as an excitation source. Time-resolved fluorescence decays were collected in the range of 305–417 nm with a step of 4 nm; time resolution was 0.027 ns/channel. The instrument response function (IRF) was measured using a Ludox solution, detecting the emission at the excitation wavelength.

To recover the fluorescence decay parameters (amplitudes and lifetimes), the global analysis approach was performed using the deconvolution DAS6 software (Horiba Jobin Yvon, USA). The set of decays collected for each sample was modeled by iterative reconvolution with three exponential components (see [12] for details). As a result, the time-resolved fluorescence decay at the wavelength l was described as:

where  is the lifetime and  is the amplitude of the i-component.

Reviewer 2:  In Figure 2, I did not understand the meaning of the statement ‘Structural elements formed at different stages of protein folding are displayed in different colors’; the authors should expand on the same; which color corresponds to what stage?.

Our response:     Thank you, we have changed the figure caption.

Reviewer 3 Report

Dear Authors,

The paper that was sent to me for review is written according to the standards for biological papers, while I am a physicist. It would be unfair to apply standards of Physics to review a Biology paper, I think.

I hope another reviewer will give your paper a proper consideration.

Sincerely,

Reviewer

Author Response

Remark:

Thank you for your submission. We noticed that the article contains western blot experiment. Our editorial office requires the authors to provide the uncropped, untouched, full original images of western blots. Would you please provide full original images of western blot to me via email? Thanks for your cooperation

Our response:      Thank you for your comment. Figure 7 is not a western blot. This is a conventional SDS electrophoresis stained with Coomassie Blue G250. Such electrophoresis allows us to analyze all protein peptides in the solution (in contrast to the western blot method). We’ve added a chapter to Materials and Methods and changed the figure caption.

Round 2

Reviewer 1 Report

The revised manuscript may be accepted.

Author Response

Thank you so much, your comments helped make the article bette

Reviewer 2 Report

The authors have answered most of the queries satisfactorily. A few minor comments below before the manuscript can be published. I do not need to see the manuscript again.

1. I still do not understand why the authors are sticking to the Boltzmann equation for fitting the urea denatured curves.  There is no way one can get Pstart or Pend unless a baseline fitting of folded and unfolded regions are carried out. This is why I had suggested the LEM method. Moreover, if it is known that the folding to unfolding transition goes through intermediates (page 7), then what is the utility of figuring out the [Urea]m, as this is not a two-state transition.

2. Page 14: The authors mention that even at pH 2.7 there is sufficient amount of structure. However the fact that the ellipticity is maximum (negative) at around 204 nm, shows that the proteins are extensively disordered. In other words, at pH 2.7, the protein(s) is essentially unfolded.

Author Response

Thanks for the comments

We have changed the structure of the article, now it meets the requirements of IJMS

We have changed the numbering of references.

We have taken into account the comments of the reviewer2 - figure 3 changed.

Reviewer 2

The authors have answered most of the queries satisfactorily. A few minor comments below before the manuscript can be published. I do not need to see the manuscript again.

  1. I still do not understand why the authors are sticking to the Boltzmann equation for fitting the urea denatured curves.  There is no way one can get Pstart or Pend unless a baseline fitting of folded and unfolded regions are carried out. This is why I had suggested the LEM method.

Our response:      

We used [Urea]m (obtained from Boltzmann equation) to estimate the influence of mutations on the separate stages of carbonic anhydrase unfolding and to compare the results with that obtained by kinetic methods [12. Nemtseva, et al. Methods Appl. Fluoresc. 2018]. However, we agree that taking into account the slopes of Pstart and Pend is necessary for a more accurate calculation of the transition parameters from the sigmoid-like curve. So, we recalculated our data using linear fitting of Pstart and Pend before retrieving [Urea]m with the Boltzmann equation (see changes in Methods chapter and in Fig. 3). We obtained almost all previous values of midpoints to be slightly shifted by 0.1-0.2 M to the higher urea concentrations. Although, the relative positions of the midpoints did not change and the idea about different midpoints for Tau1 and Tau2 dependencies is still supported (Fig. 3). We agree that the application of this approach to “not two-state” proteins is not entirely correct, but we perceive it as a necessary measure for evaluation and comparison of the wild-type protein and its mutant forms.

Reviewer 2

Moreover, if it is known that the folding to unfolding transition goes through intermediates (page 7), then what is the utility of figuring out the [Urea]m, as this is not a two-state transition.

Our response:      

We often come across the fact that readers (and reviewers) want some simple assessment of protein destabilization. We calculated the midpoints of the transitions for this purpose only. Since in Figure 3 mutants of the same protein (not different proteins with different intermediate states), the effect of mutation on the protein can be compared by this parameter.

Reviewer 2

  1. Page 14: The authors mention that even at pH 2.7 there is sufficient amount of structure. However the fact that the ellipticity is maximum (negative) at around 204 nm, shows that the proteins are extensively disordered. In other words, at pH 2.7, the protein(s) is essentially unfolded.

Our response:      

The protein is denatured, but not unfolded!

Ellipticity in a fully unfolded protein, in the region of 220 nm, should be close to zero.

The fact that there is a large ellipticity at a wavelength at 220nm means that the protein has a lot of secondary structure, which can be influenced by mutations. Ultimately, this is the influence we are investigating.